METHODS

# SpatialKNifeY (SKNY): Extending from spatial domain to surrounding area to identify microenvironment features with single-cell spatial omics data

**Shunsuke A. Sakai**[1,2,3], **Ryosuke Nomura**[1,4], **Satoi Nagasawa**[4,5], **SungGi Chi**[1],
**Ayako Suzuki**[4], **Yutaka Suzuki**[4], **Mitsuho Imai**[6,7], **Yoshiaki Nakamura**[6,8], **Takayuki Yoshino**[6,8],
**Shumpei Ishikawa**[9,10], **Katsuya Tsuchihara**[1,2], **Shun-Ichiro Kageyama**[3,11]*, **Riu Yamashita**[1,4]*

1 Division of Translational Informatics, Exploratory Oncology Research & Clinical Trial Center, National Cancer Center, Kashiwa, Chiba, Japan, 2 Department of Integrated Biosciences, Graduate School of Frontier Sciences, The University of Tokyo, Kashiwa, Chiba, Japan, 3 Department of Radiation Oncology, National Cancer Center Hospital East, Kashiwa, Chiba, Japan, 4 Department of Computational Biology and Medical Sciences, Graduate School of Frontier Sciences, The University of Tokyo, Kashiwa, Chiba, Japan, 5 Department of Breast Surgery, National Cancer Center Hospital East, Kashiwa, Chiba, Japan, 6 Translational Research Support Office, National Cancer Center Hospital East, Chiba, Japan, 7 Department of Genetic Medicine and Services, National Cancer Center Hospital East, Chiba, Japan, 8 Department of Gastroenterology and Gastrointestinal Oncology, National Cancer Center Hospital East, Chiba, Japan, 9 Department of Preventive Medicine, Graduate School of Medicine, The University of Tokyo, Bunkyo-ku, Tokyo, Japan, 10 Division of Pathology, National Cancer Center Exploratory Oncology Research & Clinical Trial Center, Kashiwa, Chiba, Japan, 11 Division of Radiation Oncology and Particle Therapy, Exploratory Oncology Research & Clinical Trial Center, National Cancer Center, Kashiwa, Chiba, Japan

* riuyamas@east.ncc.go.jp (RY); skageyam@east.ncc.go.jp (SIK)

## Abstract

Single-cell spatial omics analysis requires consideration of biological functions and mechanisms in a microenvironment. However, microenvironment analysis using bioinformatic methods is limited by the need to detect histological morphology and extend it to the surrounding area. In this study, we developed SpatialKNifeY (SKNY), an image-processing-based toolkit that detects spatial domains that potentially reflect histology and extends these domains to the microenvironment. Using spatial transcriptomic data from breast cancer, we applied the SKNY algorithm to identify tumor spatial domains, followed by clustering of the domains, trajectory estimation, and spatial extension to the tumor microenvironment (TME). The results of the trajectory estimation were consistent with the known mechanisms of cancer progression. We observed tumor vascularization and immunodeficiency at mid- and late-stage progression in TME. Furthermore, we applied the SKNY to integrate and cluster the spatial domains of 14 patients with metastatic colorectal cancer, and the clusters were divided based on the TME characteristics. In conclusion, the SKNY facilitates the determination of the functions and mechanisms in the microenvironment and cataloguing of the features.

**Data availability statement:** The code used in this study has been deposited in the documentation of the SKNY library (https://skny.readthedocs.io/en/latest/notebooks/SKNY_paper_Sakai_et_al.html).

**Funding:** SAS was supported by the Graduate School of Frontier Sciences, The University of Tokyo, through the Challenging New Area Doctoral Research Grant (Project No. C2414). The funders had no role in study design, data collection and analysis, decision to publish, or preparation of the manuscript. https://gsfs-eso.edu.k.u-tokyo.ac.jp/project/challenge-fund

**Competing interests:** I have read the journal's policy and the authors of this manuscript have the following competing interests: M.I. reports advisory role from Sumitomo Co., Ltd., Exact Sciences Corporation.; speakers' bureau from Chugai Pharmaceutical Co., Ltd.; research funding from Merck Sharp and Dohme, Inc. Y.N. reports advisory role from Guardant Health Pte Ltd., Natera Inc., Roche Ltd., Seagen Inc., Premo Partners Inc., Daiichi Sankyo Co. Ltd., Takeda Pharmaceutical Co. Ltd., Exact Sciences Corporation, and Gilead Sciences Inc.; speakers' bureau from Guardant Health Pte Ltd., MSD K.K., Eisai Co. Ltd., Zeria Pharmaceutical Co. Ltd., Miyarisan Pharmaceutical Co. Ltd., Merck Biopharma Co. Ltd., CareNet Inc., Hisamitsu Pharmaceutical Co. Inc., Taiho Pharmaceutical Co. Ltd., Daiichi Sankyo Co. Ltd., Chugai Pharmaceutical Co. Ltd., Becton, Dickinson and Company, and Guardant Health Japan Corp; research funding from Seagen Inc., Genomedia Inc., Guardant Health AMEA, Inc., Guardant Health, Inc., Tempus Labs Inc., Roche Diagnostics K.K., Daiichi Sankyo Co. Ltd., and Chugai Pharmaceutical Co. Ltd. T.Y. reports honoraria from Chugai Pharmaceutical Co. Ltd., Takeda Pharmaceutical Co. Ltd., Merck, Bayer Yakuhin, Ono Pharmaceutical, and MSD K.K; consulting fee from Sumitomo Corp., and research grant from Amgen, Chugai Pharmaceutical Co., Ltd., Daiichi Sankyo Co., Ltd., Eisai, FALCO Biosystems, Genomedia Inc., Molecular Health, MSD, Nippon Boehringer Ingelheim, Ono, Pfizer, Roche Diagnostics, Sanofi, Sysmex, and Taiho Pharmaceutical Co. Ltd., outside the submitted work. The other authors declare no competing interest.

## Author summary

The advent of high-resolution and high-density spatial omics platforms has created a growing need for practical analytical tools in cancer research. While significant efforts have been made to develop unsupervised clustering methods, advancements in downstream analyses have been relatively slower. To address this issue, we developed SpatialKNifeY (SKNY), a versatile toolkit designed to analyze spatial omics data by defining the spatial domains of cancer cells and their microenvironment. SKNY offers a suite of analyses, including clustering and trajectory analysis, with a unique capability to extract spatial domains and their surrounding regions. The tool enables integrated studies of cancer cells alongside their stroma, immune cells, and vascular environment. Using SKNY, we quantified the vascular and immune environments surrounding cancer cells during progression, revealing insights consistent with established cancer pathology and progression models. These results highlight the toolkit's utility and the biological interpretability of its analyses, providing a valuable resource for spatial omics research.

## Introduction

Single-cell spatial omics platforms, such as Xenium, CosMx [1], and PhenoCycler [2], offer opportunities for the investigation of hundreds or thousands of genes in various organs and tissue types. The resolutions of the methods are at the single-cell level, providing deep insights into the localization of the expression of multiple genes in a particular microenvironment, which includes not only cancer cells but also immune cells and non-immune stromal cells. A key consideration in microenvironment analysis is the integration of gene expression and histological features to obtain a comprehensive understanding of biological functions and mechanisms. Classical methods that examine the tumor microenvironment (TME) using a microscope capture histological features through staining or fluorescence-based technologies, leading to the discovery of pathological mechanisms in the microenvironment [3]. However, in the current omics era, with the large number of specimens and gene panels, manual physical approaches are inefficient and impractical.

To address the high throughput of omics data, several third-party tools, such as Seurat and Scanpy, have been developed to efficiently analyze expression data from thousands of gene panels and samples [4–10]. Methods inherited from single-cell RNA-seq have been implemented, including cell clustering [11–14], trajectory analysis [15–18], and ligand-receptor analysis [19–21]. These analytical methods use gene expression but do not consider molecular or cellular location. Hence, the integration of gene expression and location information is necessary for optimizing spatial omics analysis of the microenvironment.

In response to the demand, several tools dedicated to spatial omics have been developed, such as clustering analyses that integrate positional information with gene expression [22] and ligand-receptor enrichment analysis at each spot in a space partitioned on a grid [23]. Although the methods are attractive for application in analyzing spatial information, micro-environmental analysis is limited by the lack of direct histological information. Recently, the STAGATE algorithm [24] has been developed for detecting spatial domains (i.e., regions with similar spatial expression patterns), and Sopa [25] was constructed to extend 'spatial domain' analysis to single-cell spatial omics data. The methods can detect spatial domains that reflect and functionally resemble tumor, stromal, and vascular histologies. Even though these tools

are invaluable for extracting and characterizing spatial domains, they are limited in analyzing a particular spatial domain's surrounding area, the microenvironment.

Here, we extended the concept of the spatial domain to the microenvironment, which encompasses inside, peri-, and outside sections of the spatial domain, with the aim of estimating the functions and mechanisms of the microenvironment (Fig 1A). We developed an image processing-based toolkit, SpatialKNifeY (SKNY), to analyze the spatial domains in spatial omics data (Output 1-3) and extend it to the microenvironment (Output 4) (Fig 1B). Single-cell spatial transcriptomics data from Xenium [26] were used to detect spatial domains of tumor for analyzing the TME (Output 1: *Detection*) (Fig 1C). Clustering of the spatial domains resulted in the formation of clusters consistent with malignancy and subtypes (Output 2: *Clustering*), and the trajectory among spatial domains was estimated to represent the tumor progression process (Output 3: *Trajectory estimation*). The analysis extended from the spatial domain into the TME and assessed infiltration of endothelial cells into the tumor (Output 4: *Spatial stratification*). Moreover, to conduct an integrated analysis with multiple samples, SKNY was applied to a Xenium dataset of 14 patients with metastatic colorectal cancer. The results suggest that the SKNY can provide microenvironment analysis and may provide essential insights into their pathological functions. The SKNY algorithm is available under an open-source license (https://github.com/shusakai/skny).

## Results

### SKNY detects tumor spatial domains from Xenium breast cancer data

In the present study, to detect the spatial domain with the SKNY, Xenium breast cancer data from a previous report were used [26]. A hematoxylin and eosin (HE)-stained image of the specimen from a previous report is shown (Fig 2A). The specimen on a single slide contained various tumor tissues, including ductal carcinoma in situ (DCIS) and invasive ductal carcinoma (IDC). Using Xenium data, the SKNY algorithm was applied to detect tumor spatial domains (yellow) and extract their boundaries (green) based on the expression levels of the epithelial cell marker *CDH1* (Fig 2B). Independently, the STAGATE algorithm [24] was used to identify tumor spatial domains (S1A, S1B, and S1C Fig), resulting in high concordance with the SKNY results (Jaccard similarity coefficient = 0.85). The results suggest that the image-processing-based spatial domain extraction of the SKNY method is consistent with previous methods. Moreover, spatial domains were extracted from ovarian cancer, colorectal cancer, and melanoma data and were found to be visually consistent with the HE-stained images (S2 Fig). The inward/outward areas from the extracted spatial domain boundaries were measured (S3A Fig), and the contour line was delineated at 30-μm intervals to spatially stratify the TME (Figs 2C and S3B). High-power field images, including single (Fig 2C Left), triple (Fig 2C Middle), and multiple spatial domains (Fig 2C Right), showed visual concordance between the spatial domains and HE staining images for tumor detection.

To confirm that the spatial domains were partitioned correctly between the tumor and stroma, the expression levels of several marker genes were examined in the stratified (−90, −60] to (+120, +150] sections in the total field. The results showed that cancer cell marker genes, such as *CDH1*, *EPCAM*, *FOXA1*, and *GATA3,* were enriched within the spatial domain (sections (−120, −90], (−90, −60], (−60, −30] and (−30, 0]) (Fig 2D). The myoepithelial cell marker genes, such as *KRT5*, *KRT14*, *MYLK*, and *ACTA2*, were enriched around the spatial domain boundary (the section of (0, +30]), and the macrophage, lymphocyte, endothelial cell, and stromal cell markers, such as *CD68*, *TRAC*, *PECAM1*, and *MMP2*, respectively, were enriched on the outside (the sections of (+30, +60], (+60, +90], (+90, +120], and (+120, +150]). The spatial localization of gene expression showed that *EPCAM* was overrepresented

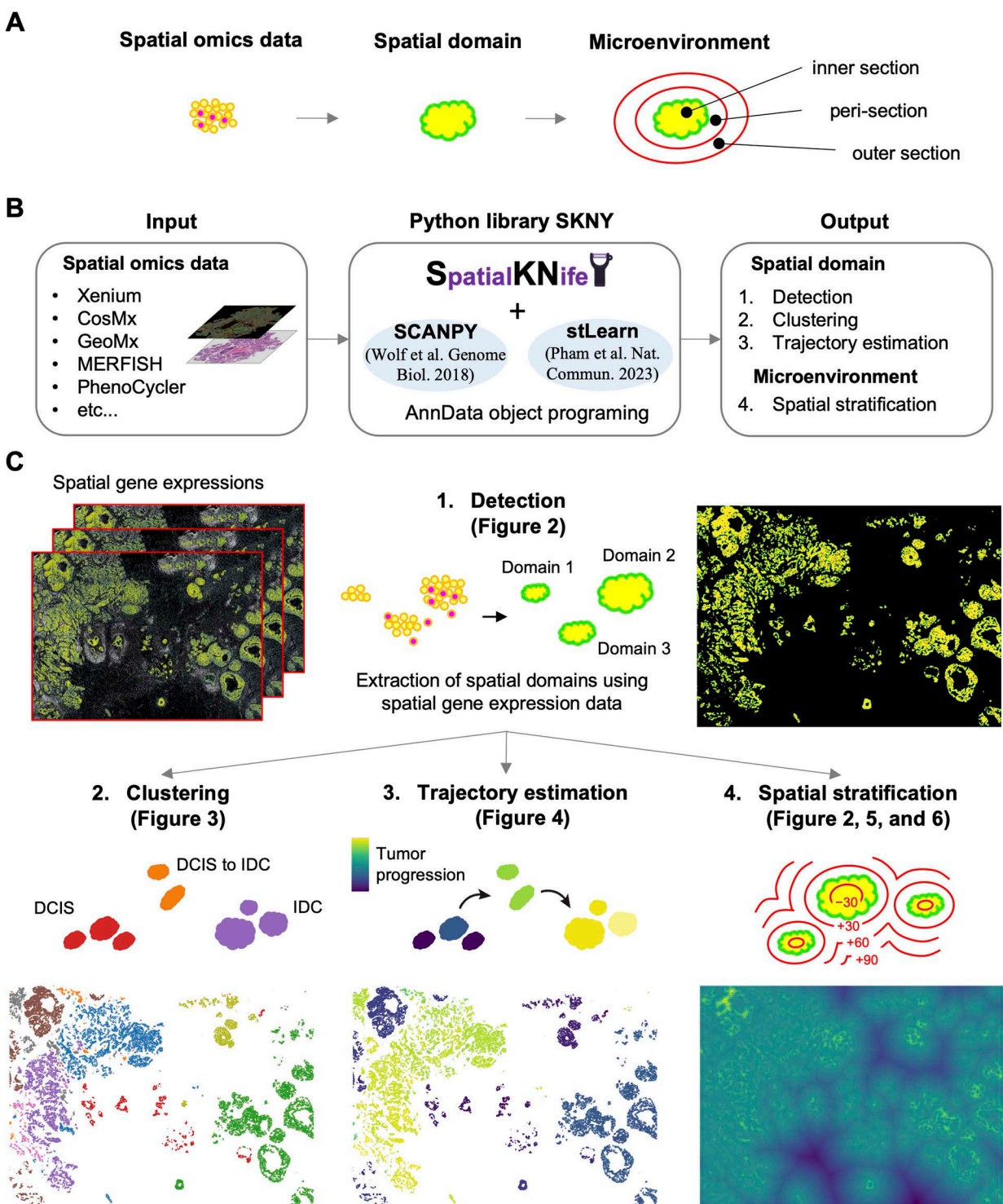

**Fig 1. SpatialKNifeY analysis landscape.** (A) The concept of the extension from spatial omics data and spatial domain to the microenvironment. (B) Implementation of SpatialKNifeY (SKNY). A Python library of SKNY depends on stlearn [23] and scanpy [9] functions (see "Methods") and AnnData object programming [10]. (C) Outputs from SKNY analysis. Detection (Output 1, see "Fig 2") delineates spatial domains based on a user's positive and negative marker gene expressions. Clustering (Output 2, see "Fig 3") makes clusters of spatial domain units based on the mean expression of each gene. Trajectory estimation (Output 3, see "Fig 4") refers to the trajectory among spatial domains and pseudotime. Spatial stratification (Output 4, see "Fig 2", "Fig 5", and "Fig 6") measures the distance from tumor boundary to each coordinate on the space and makes contour lines based on the distance.

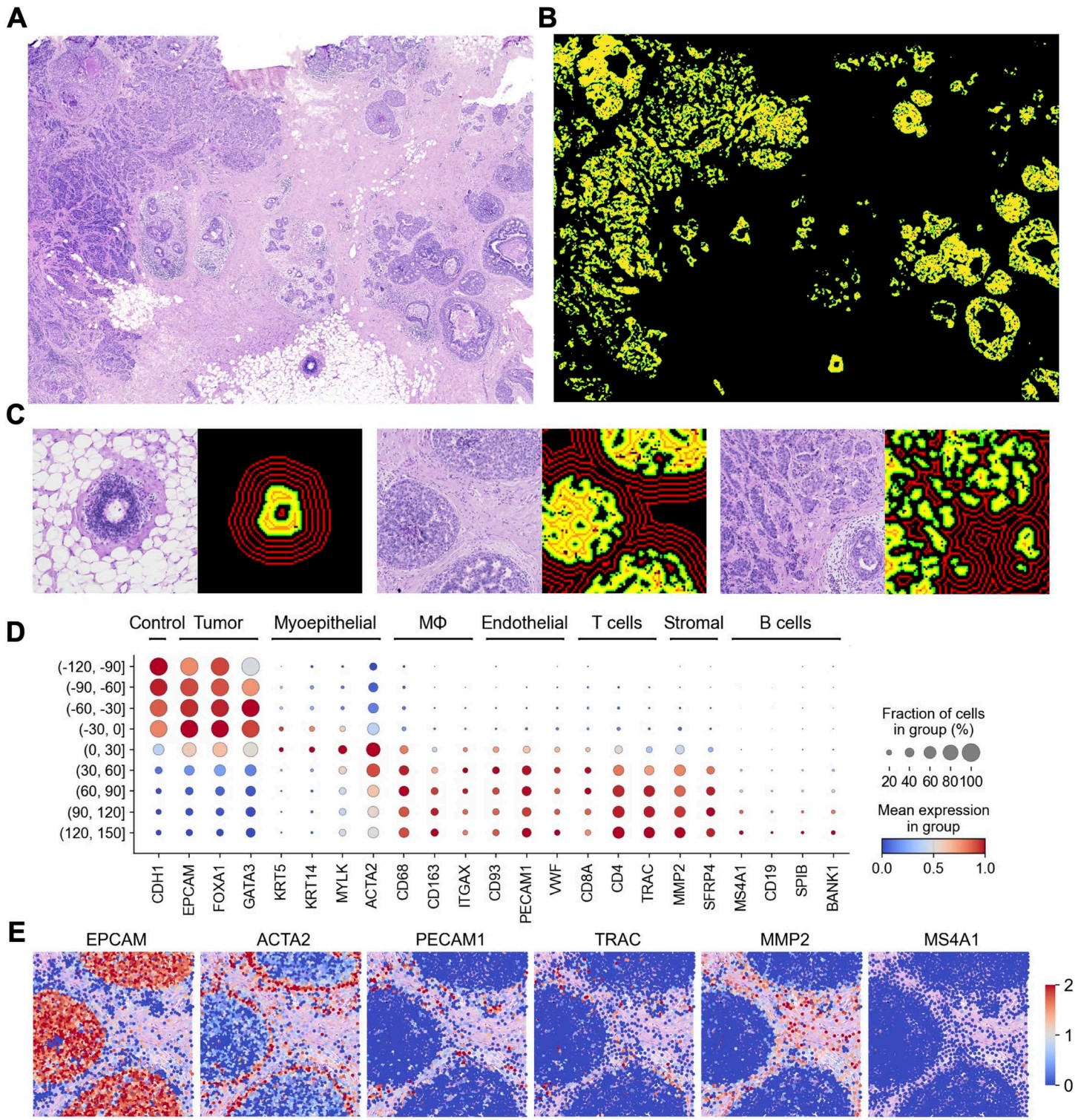

**Fig 2. Detection of spatial domain using Xenium data accurately discriminates between the tumor and stromal region.** (A) H&E staining image of breast cancer. (B) Detected spatial domains. The yellow and green colors indicate spatial domains and the boundary, respectively. (C) H&E staining images and spatial domain(s) from three ROIs. The red contour lines indicate distance from the surface of spatial domains at the 30-μm interval. (D) Dotplot showing marker genes of each cell type. The color bar indicates the scaled mean count, and the size indicates the percentages of the gene expressions. (E) Spatial expression distribution of cell marker genes in the ROI. The color bar indicates the scaled mean count.

within the spatial domain, *ACTA2* at the boundary, and *PECAM1*, *TRAC*, and *MMP* outside the domain (Fig 2E). The results suggest that the spatial domains stratified using the SKNY algorithm can be divided into tumors, peritumor, and stroma.

## SKNY clusters the spatial domains with multiple mixed cell types into subclusters using the UMAP algorithm

Next, to assess the diversity of cells within extracted spatial domains, the α-diversity index (Chao1) was compared based on the gene expression between cancer cells and spatial domains. The results indicated that gene expression in the spatial domain was significantly more diverse than that in the cancer cells (*P*<0.001) (S4A Fig), suggesting that the spatial domains contain various cells, not only cancer cells. A similar trend was confirmed between the various cells independently annotated by Janesick *et al.* [26] and the spatial domain by the three α-diversity indices of Chao1, observed features (the number of unique genes detected in each cell or spatial domain), and Shannon (S4B-S4D Fig). Moreover, diversity variance of Chao1 was greater in spatial domains (standard deviation [SD]=62.1) than in cancer cells (SD=34.2). Hence, we hypothesized that the heterogeneity among spatial domains originated not only from cancer cells but also from diverse cells in the microenvironment. Here, we performed clustering of spatial domains to evaluate heterogeneity among spatial domain

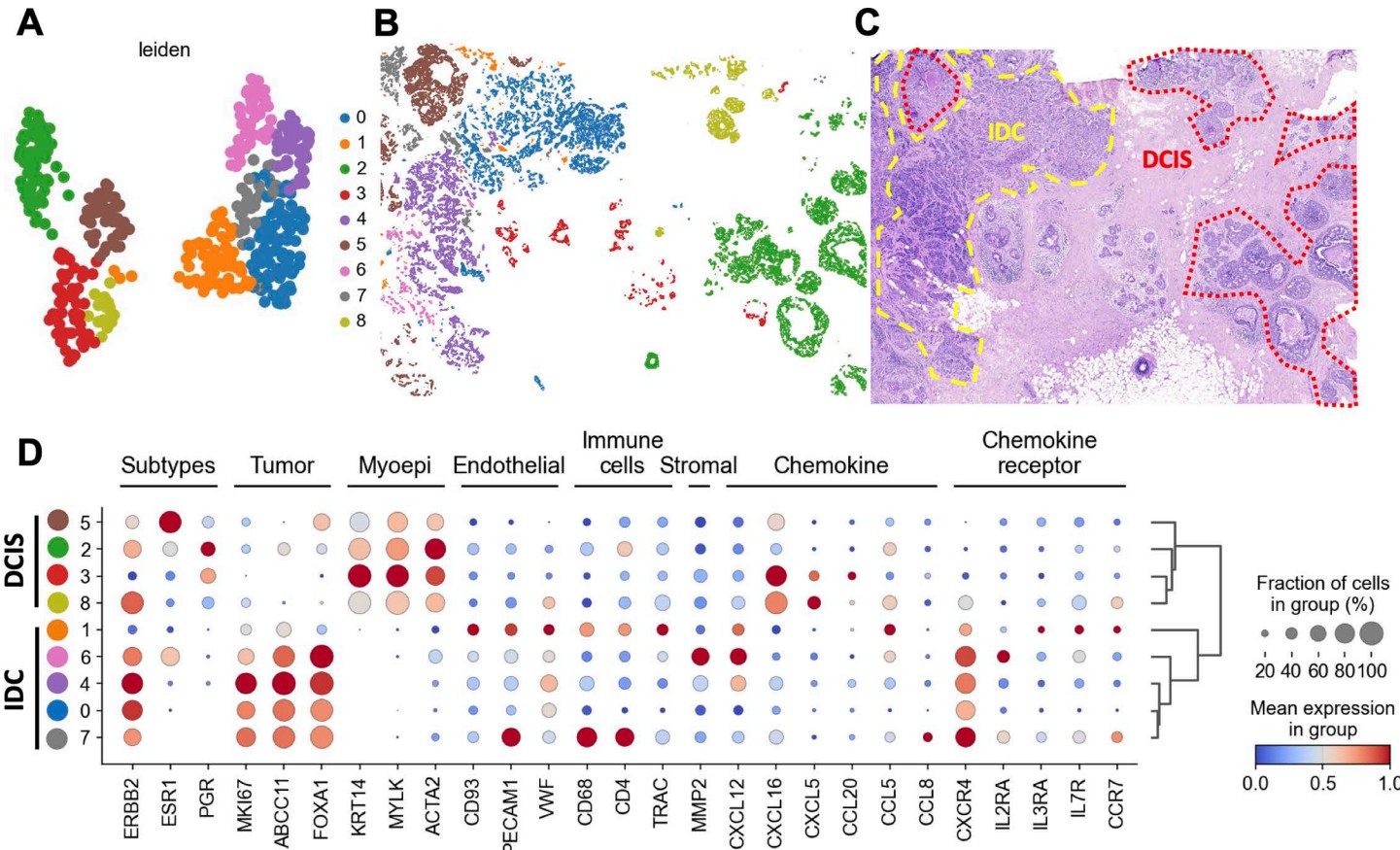

**Fig 3. Clustering and annotation of spatial domain based on gene expression.** (A) Two-dimensional plot based on UMAP loadings of gene expression of spatial domains. The colors indicate clusters. (B) Spatial distribution of each cluster. (C) H&E image with histological annotations. (D) Dotplot showing markers of cell types and expression patterns of genes associated with tumor subtypes.

microenvironments. The gene expression data (313 genes) were reduced dimensionally by principal component analysis (PCA), resulting in nine clusters (0–8) based on their similarity in PCA space. Each spatial domain was placed in the two-dimensional space using UMAP (Fig 3A) and the original space (Fig 3B). To annotate the clusters with histology, we showed HE staining images based on the previous report (Fig 3C). Combining this histology on HE staining with the clusters shown in Fig 3B, we found that clusters 2, 3, 5, and 8 corresponded to non-invasive ductal carcinoma in situ (DCIS), whereas clusters 0, 1, 4, 6, and 7 corresponded to invasive ductal carcinoma in situ (IDC).

To provide detailed annotations of each spatial domain cluster, we examined the expression of several marker genes. In clusters 0, 1, 4, 6, and 7 (IDC clusters), *MKI67* and *ERBB2* were expressed highly. Conversely, in clusters 2, 3, 5, and 8 (DCIS clusters), the myoepithelial cell markers *ACTA2*, *MYLK*, and *KRT14* were expressed highly. The results suggest that gene expression in each spatial domain was consistent with the histological annotation (Fig 3D). Interestingly, cluster 1 showed high expression of endothelial cell markers, including *PECAM1*, *VWF*, and *CD93*, as well as chemokines and chemokine receptor genes associated with cell migration, *CXCL12* and *CXCR4*. Furthermore, *MKI67*, *ABCC11,* and *FOXA1* expression were moderate in cluster 1 compared to those in other IDC clusters (Fig 3D). Considering the moderate expression of the cancer-associated genes and their midpoint in the UMAP space (Fig 3A), Cluster 1 may represent a spatial domain at an intermediate stage in the transition from DCIS to IDC.

## SKNY estimates spatial domain trajectory, which reflects tumor progression

To estimate the spatial domain trajectory from DCIS to IDC, a partition-based graph abstraction (PAGA) algorithm [15] was used to construct an adjacency graph representing the topology of expression patterns for each cluster (Fig 4A). The adjacency graph is divided into clusters 2, 3, 5, and 8 (DCIS) and clusters 0, 4, 6, and 7 (IDC), where cluster 1 connects the DCIS and IDC clusters. Additionally, cluster 3, exhibiting the lowest tumor marker gene expression, as shown in Fig 3D, was located at the lower end. This structure is consistent with the hypothesis that the spatial domain of DCIS clusters transitions to the IDC cluster via cluster 1. Moreover, we confirmed that other algorithms, such as Monocle [27] and Slingshot [17], estimated a similar trajectory (S5A and S5B Fig). The pseudotime with cluster 3 as the root was determined and placed in the two-dimensional space of the PAGA algorithm and the original space (Fig 4B Left and 4C). We evaluated the correlation between the pseudotime and *MKI67* ($r$=0.52, $P$<0.001, Pearson coefficient)/*ACTA2* ($r$=−0.47, $P$<0.001). The pseudotime illustrated tumor progression (Fig 4B Middle and Right).

To identify characteristic gene expression at points on this pseudotime axis, we hypothesized three tumor progression paths (cluster 3→8→1→7→4: IDC path #1, 3→5→1→7→4: IDC path #2, and cluster 3→2: DCIS path) and evaluated trends in gene expression along the paths. In IDC paths #1 and #2, the expression of myoepithelial cell markers (*ACTG2* and *MYLK*) tended to decrease in the early stages of progression, whereas that of malignant markers (*ERBB2*) tended to increase in the later stages (Fig 4D). In contrast, the myoepithelial cell and malignant marker fluctuations appeared to be moderate in the DCIS path. Moreover, in IDC paths #1 and #2, marker genes for endothelial cells (*VWF* and *PECAM1*), lymphocytes (*CD4*), macrophages (*CD68*), chemokines (*CXCL12* and *CCL5*), and chemokine receptors (*CXCR4*) were expressed highly at the intermediate stages of cancer progression. Similarly, in the DCIS path, *CD4*, *CD68*, and *CCL5* showed increased expression with progression. The findings suggest that endothelial cells and chemokine signaling are involved in the transition from DCIS to IDC. We also examined the spatial distribution of gene expression within the region of interest (ROI)

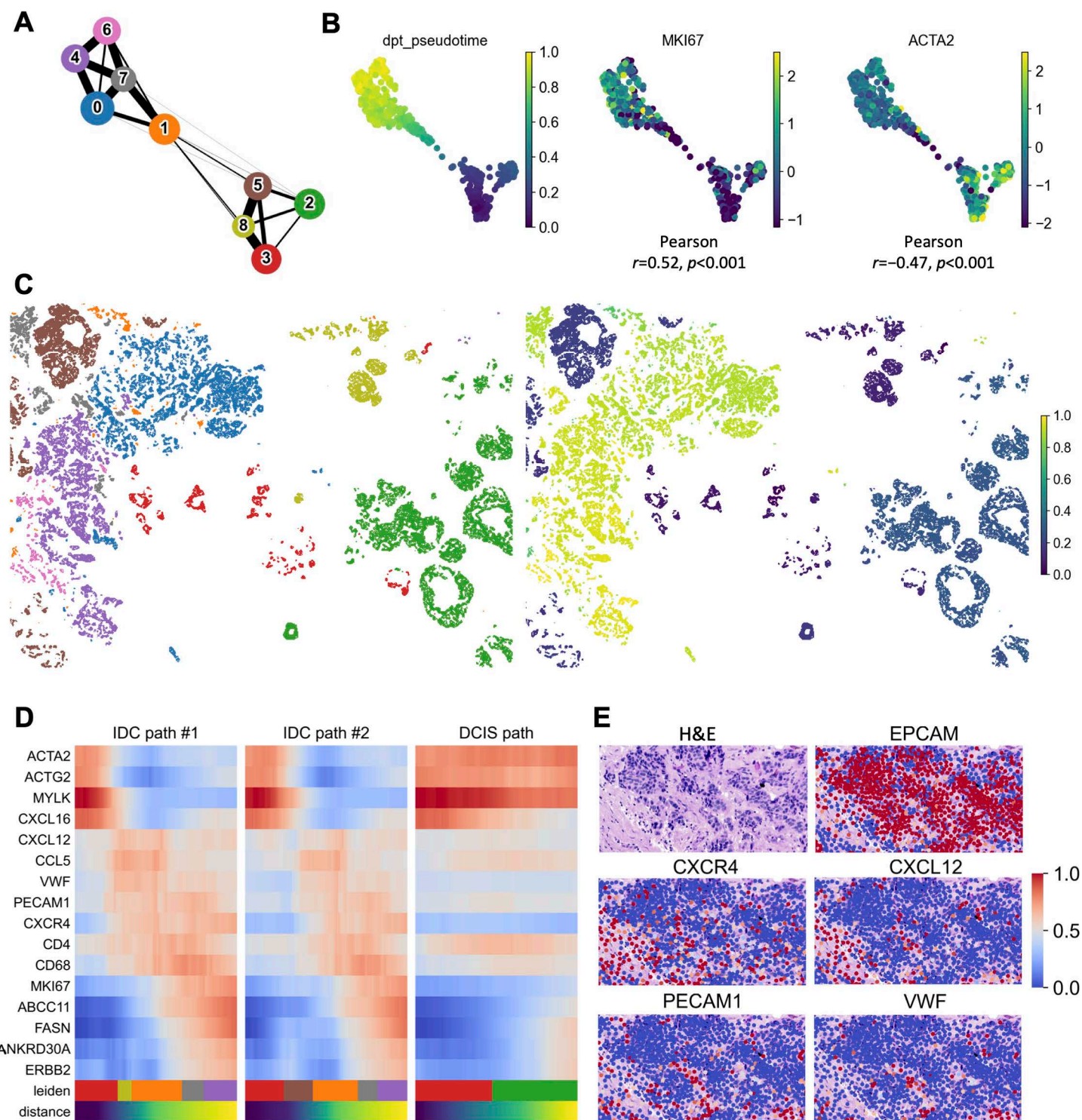

**Fig 4. Estimating spatial domain trajectory reveals temporal gene expression gradient along cancer progression.** (A) PAGA graph constructed based on the expression data of the spatial domains. (B) PAGA-initialized spatial domain embeddings with estimated pseudotimes, *MKI67*, and *ACTA2* expressions. Pearson's correlation coefficients and P values were used to evaluate linear relationship between pseudotimes and scaled expression of *MKI67*/*ACTA2*. (C) Spatial distribution of clusters and pseudotimes. (D) Heatmap showing gene expression level along with pseudotimes on three progression paths. (E) Representative HE staining images and gene expression on the ROI. Color bar indicates the scaled mean count.

corresponding to the transition phase from DCIS to IDC. The results showed a pattern in which *PECAM1*, *VWF*, *CXCR4*, and *CXCL12* appeared to infiltrate regions of the tumor delineated by HE staining and *EPCAM* (Fig 4E). This also suggests that during the transition from DCIS to IDC, endothelial cells may infiltrate tumors and activate chemokine signals.

### SKNY quantifies the infiltrating of endothelial cells to spatial domains in the microenvironment

We extended the spatial domains with their expression into their inner, peri-, and outer sections, namely, microenvironments, to quantitatively compare endothelial cell infiltration into tumors. We stratified the distance from the boundary of the spatial domain into 30-µm sections and extracted (−30, 0] (inner), (0, +30] (peri-), and (+30, +60] (outer) sections of each cluster (Fig 5A). Although no significant differences in expression levels were observed in the (+30, +60] section, significant differences among clusters were observed in the (−30, 0] section for endothelial cell markers *PECAM1* and *VWF* ($P=0.0053$ and $< 0.001$, Kruskal−Wallis test, respectively), with relatively high expression in cluster 1 (Fig 5B). The spatial autocorrelation coefficient (Geary's *C*) of *PECAM1*, *VWF*, and *CD93* were calculated within each cluster, revealing that the DCIS-to-IDC cluster (Cluster 1) tended to have a lower score indicating higher autocorrelation (S5C Fig). To confirm the spatial expression patterns, ROIs selected from clusters 3, 8, 1, and 0 were extracted, and the distribution of cancer cell (*EPCAM* and *CDH1*) and endothelial cell (*VWF*, *PECAM*, *CD93*) markers was examined using Xenium Explorer. In clusters 3 and 8 (DCIS cluster), endothelial cell markers were localized outside the spatial domain, whereas in cluster 1 (DCIS-to-IDC cluster), the markers were localized in the tumor spatial domain (Fig 5C). Moreover, cluster 0 (IDC cluster) appeared to remain in the gaps where the cancer cells had migrated (Fig 5C Right). The results demonstrate that the analysis, expanded from the spatial domain to the microenvironment, could reflect the infiltration of endothelial cells into the tumor.

To analyze other stromal cells, we examined changes over the pseudotime (IDC path #1) in the expression of endothelial cells (*PECAM1*), macrophages (*CD68*), matrix metalloproteinases (*MMP2*), chemokine receptors (*CXCR4*), and chemokines (*CXCL12*) in each TME section, at (−30, 0] (inner), (0, +30] (peri-), and (+30, +60] (outer), respectively. In the inner section, *PECAM1* ($P=0.023$), *CD68* ($P<0.001$), and *CXCR4* ($P<0.001$) showed increases during the transition period from DCIS to IDC (clusters 8, 1, and 7), whereas in the peri-section, *PECAM1* ($P=0.035$) and *CXCR4* ($P<0.0024$) showed an increase during (Kruskal−Wallis test, Bonferroni-corrected *P* values) (S6A Fig). In contrast, in the peri- and outer sections, *MMP2* ($P<0.001$ and $P=0.020$, respectively) showed an increase in the peaks in early DCIS (cluster 3) and late IDC (cluster 4). We summarized the temporal sequences of the expression of the genes. In the early stages of cancer progression, *MMP2* expression was upregulated in the peri- and outer regions (S6B Fig). In the tumor progression from non-invasive to invasive cancer, infiltration of endothelial cells (*PECAM1*) and macrophages (*CD68*) was noted in the tumor interior, in addition to increased chemokine signaling (*CXCR4*). After invasion, *MMP2* expression was upregulated in the peritumor and outer regions.

### SKNY quantifies the spatial localization of immune cells around spatial domains as a microenvironment

We focused on a microenvironment around the spatial domain and compared the localization of several immune cells between DCIS and IDC areas. Four ROIs were extracted for both DCIS and IDC, with sufficient inclusion of both tumor and stroma (Fig 6A). To quantify the spatial localization of gene expression outside the spatial domains, the area into was

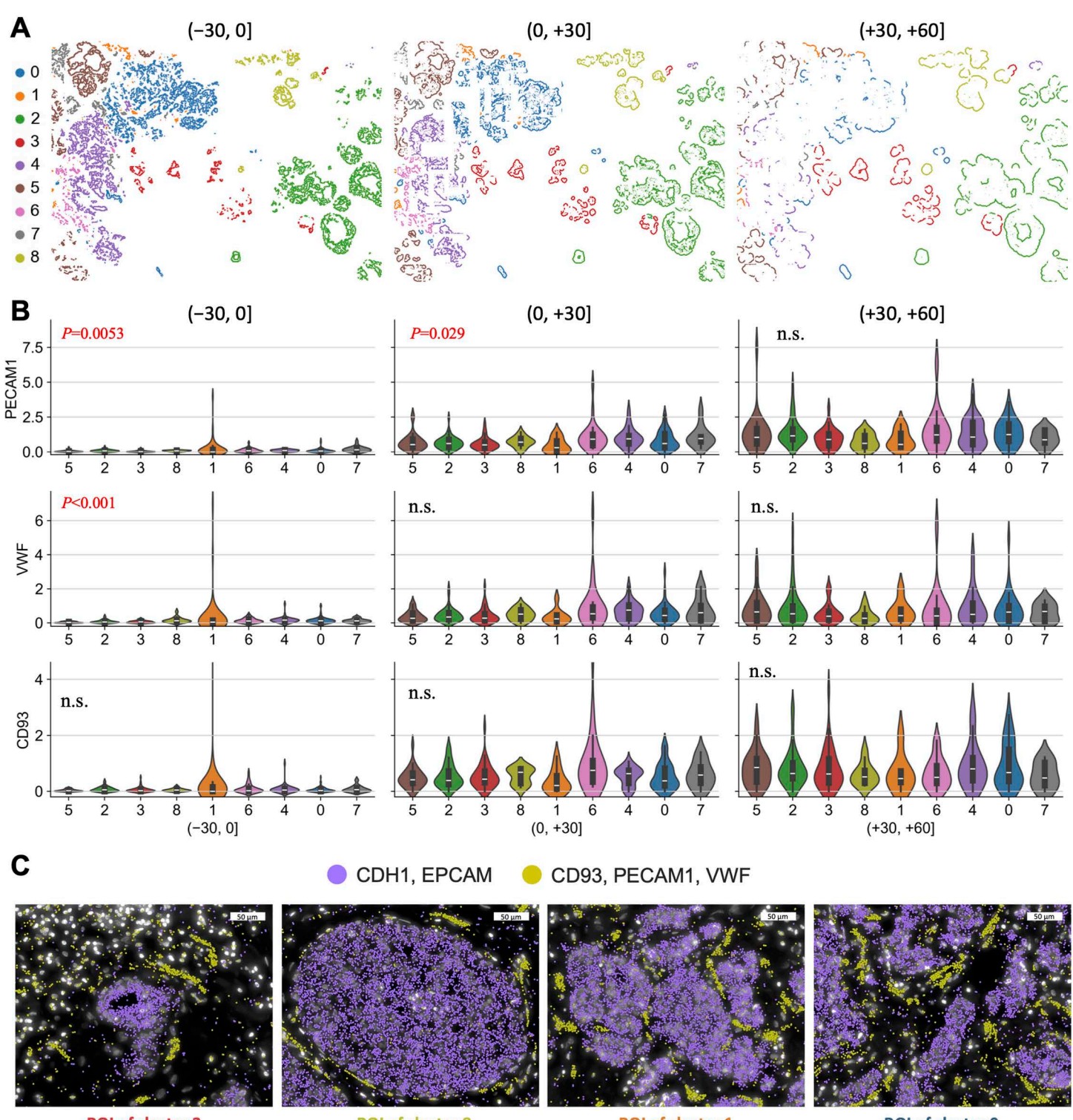

**Fig 5. Spatial stratification of each spatial domain cluster elucidating endothelial cell invasion into the tumor.** (A) Spatial distributions of stratified spatial domain clusters into (−30, 0], (0, +30], and (+30, +60] sections. (B) Violin plots showing the endothelial cell marker gene expressions (*PECAM1*, *VWF*, and *CD93*) for each cluster in the (−30, 0], (0, +30], and (+30, +60] sections. The x-axes indicate cluster numbers, and the y-axes indicate scaled gene expression levels. The annotated values are the *P* values of the significance test. (C) Representative images of DAPI with epithelial cell markers (*CDH1* and *EPCAM*) and endothelial cell marker (*CD93*, *PECAM1*, and *VWF*) expression for four ROIs.

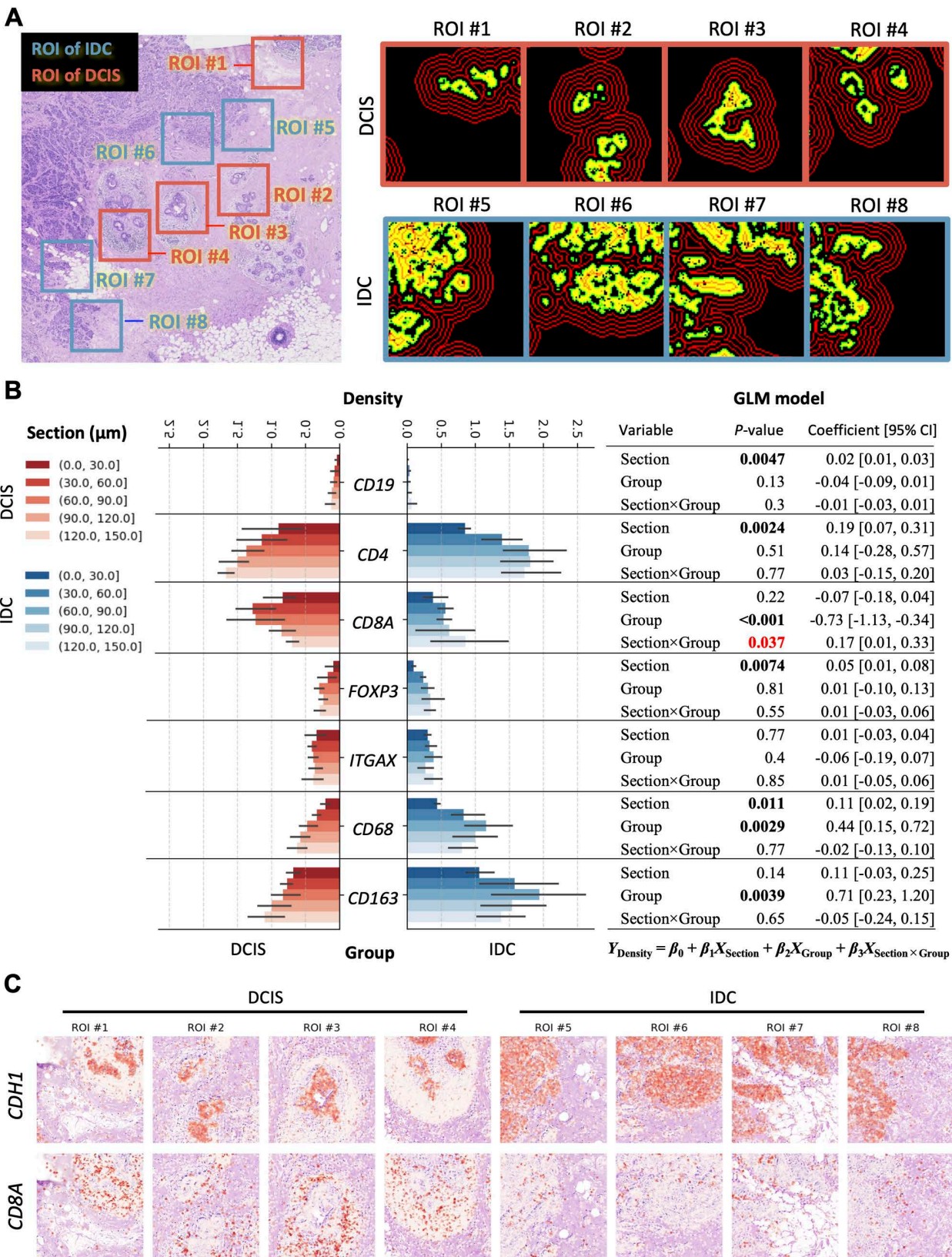

**Fig 6. Comparison of immune cell distribution in microenvironments between DCIS and IDC regions.** (A) ROIs for DCIS and IDC clusters, respectively. The ROIs in red indicate clusters for DCIS cluster and those in blue for IDC cluster. (B) Bar plot shows the expression density of

various immune cell markers stratified by section. The red series indicates DCIS, and the blue series indicates IDC. The respective color gradients indicate each stratified interval. Error bars represent 95% confidence intervals (CI). The p-values and regression coefficients [95% CI] for each constructed GLM model are shown on the right side. (C) Spatial expression distribution of CDH1 and CD8A in each ROI.

stratified into (0, +30], (+30, +60], (+60, +90], (+90, +120], and (+120, +150] sections based on the measured distance from the spatial domains' surface. We compared the expression of immune cell markers, including *CD19*, *CD4*, *CD8A*, *FOXP3*, *ITGAX*, *CD68*, and *CD163*, in the sections between DCIS and IDC areas (Fig 6B). A general linearized model (GLM) model was constructed to conduct a statistical analysis of the differences in expression of the markers among the sections. The objective variable was defined as expression density ($Y_{density}$), while the explanatory variables were section ($X_{section}$), group ($X_{group}$), and interaction term between section and group ($X_{section \times group}$). The results showed that *CD8A* was elevated markedly in *DCIS* ($P<0.001$), and *CD163* was increased significantly in IDC ($P = 0.0039$). Additionally, the GLM model suggested a significant interaction effect of *CD8A* ($P=0.037$), with a distinct spatial distribution pattern featuring peaks at (+30, +60] in DCIS and (+120, +150] in IDC. *CD8A* was concentrated closer to DCIS and farther away from IDC (Fig 6C).

To identify the molecular pathways associated with the differences in *CD8A* localization, genes that exhibited a strong correlation with *CD8A* distribution we extracted from the total 313 genes in the Xenium panel [26] for each of DCIS and IDC (S7A Fig). The correlation coefficients for *CCL5*, *CCL8*, *CD8B*, and so forth, were observed to be higher in DCIS, while those for *CX3CR1*, *KRT14*, *CD8B*, and so forth were higher in IDC. The spatial localizations of *CCL5* and *CX3CR1* were also correlated with that of *CD8A* in DCIS and IDC areas, respectively (S7B and S7C Fig). Moreover, KEGG enrichment analysis was conducted on the genes with correlation coefficients ≥ 0.75. The results indicated that genes such as "Antigen processing and presentation" (hsa04612) in DCIS and "Primary immunodeficiency" (hsa05340) in IDC were enriched significantly (S7D Fig). The results demonstrated that the environment surrounding the spatial domain undergoes alterations at the pathway level in the context of cancer progression.

## SKNY can integrate multiple samples of metastatic colorectal cancer and cataloguing features of the microenvironment

Finally, to conduct an integrated analysis of multiple samples, SKNY was applied to the Xenium dataset pertaining to metastatic colorectal cancer ($N_{sample}=24$, $N_{patient}=14$) from the TRIUMPH trial [28] (S8 Fig). A total of 2,151 spatial domains, including not only tumor cell but also non-tumor cell, were extracted and classified into 12 clusters in the tSNE space (Fig 7A). Conversely, as a conventional analysis at the cellular level, 391,639 tumor cells were extracted in isolation from non-tumor cells such as fibroblasts and immune cells (S9A-S9C Fig). The tumor cells were subsequently clustered, resulting in the formation of 15 distinct clusters within the tSNE space (Fig 7A). The patient IDs were linked to the spatial domains and tumor cells (Fig 7B). The results indicated that within the spatial domain-based space, each cluster encompassed multiple patients. In contrast, within the tumor cell-based space, each patient distinctly separated the clusters. Furthermore, spatial domain cluster 3 exhibited high *COL1A1* expression, while spatial domain cluster 5 showed elevated *FGFR2* and *CXCL12* levels. The genes are expressed by cells in the TME, playing crucial roles in immune response and treatment resistance. However, such environmental clusters could not be identified through single-cell cluster analysis of tumor cells alone (S10 Fig). The findings demonstrate that SKNY's integrated analysis of multiple samples is capable of cataloguing critical microenvironmental factors.

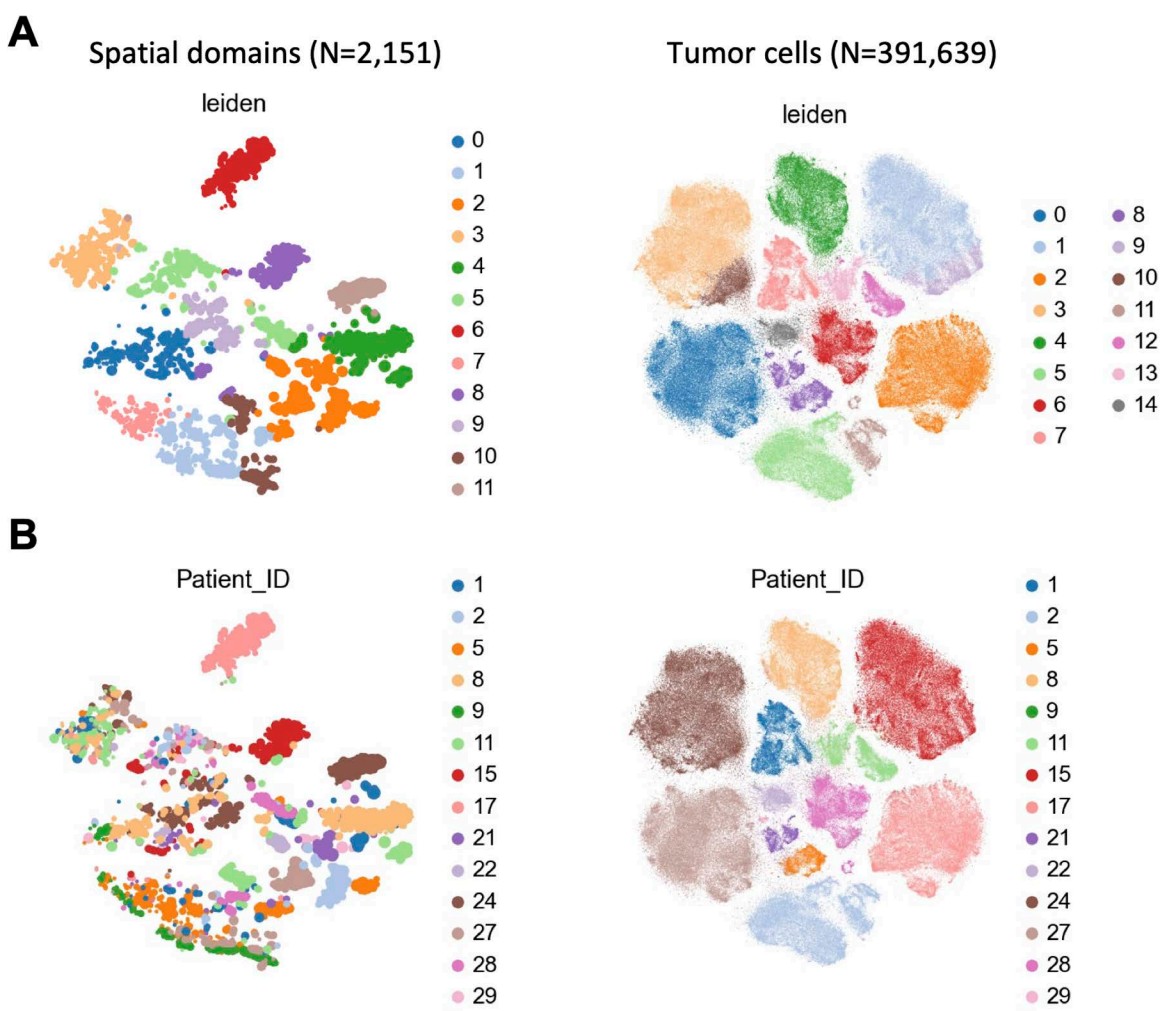

**Fig 7. Cataloguing spatial domains using Xenium data of metastatic colorectal cancer in TRIUMPH trial.** (A) t-Distributed stochastic neighbor embedding (tSNE) plot of 2,151 spatial domains (left) and 391,639 tumor cells (right) from 23 Xenium data. The plots are colored according to the clusters determined by the leiden algorithm [29]. (B) tSNE from A colored by patient ID.

## Discussion

In the present study, the SKNY algorithm was applied to spatial transcriptomics data to predict the cellular and molecular functions and mechanisms in the TME. The TME includes diverse cells, such as cancer-associated fibroblasts, stromal cells, and immune cells, which are involved in cancer progression [30], and the TME concept has also been incorporated into clinical research on breast cancer [31]. For example, immunohistochemical pathological analysis has shown that intratumoral macrophages stained by CD68 are correlated with malignancy [32,33] and that intertumoral microvessel density assessed based on CD31, which reflects angiogenesis, is a key poor prognosis factor [34,35]. In breast cancer, high *Ki67* and *HER2* expression is associated with malignancy [36], whereas destruction of myoepithelial cells is associated with tumor invasion [37]. Consistent with these previous reports on pathology, the results of *spatial stratification* (Output 4) analysis, which showed an overrepresentation of *CD68* and *PECAM*1 (*CD31*) within the spatial domain of the invasive tumor (Figs 5 and S6), demonstrated the infiltration of macrophages and endothelial cells

into malignant cancer. Moreover, *MMP2* was overexpressed in the early and late stages of tumor progression in the stromal area, and *CXCR4* and *CXCL12* were enriched after mid-stage progression inside the tumor (S6 Fig). MMPs contribute to the sprouting of vascular endothelial cells by degrading the vascular basement membrane and extracellular matrix in the early stages of tumor angiogenesis [38], and CXCR4/CXCL12 signaling pathway mediates cell migration signals and metastasis processes [39]. The results are consistent with the previous findings, suggesting that our algorithm can accurately estimate compatible biological mechanisms in the TME.

The *trajectory estimation* (Output 3) analysis was used to construct the tumor progression trajectory of the spatial domains (Fig 4). The interaction of various cells in the TME is considered crucial for cancer progression [30]; therefore, the progression trajectory should be determined by integrating all cells in the TME rather than by focusing solely on cancer cells. In our results, during the transition from DCIS to IDC, an overrepresentation of vascular endothelial cells expressing *PECAM1* and *VWF*, as well as an increase in the *CXCL12* and *CXCR4* chemokine-chemokine receptor pair, was noted. The results are consistent with the known mechanisms by which cancer cells acquire invasive potential through endothelial cells [40] and the associated induction of cell migration signals from chemokines [39]. Most importantly, gene expression from non-cancer cells was the 'missing link' between DCIS and IDC in the trajectory, suggesting the utility of the approach for integrating all cells within the spatial domain. Furthermore, our data estimated the trajectory from the root to *PGR*-positive DCIS without progression to IDC. Reduced *PGR* expression has been suggested as a surrogate marker for *GATA3* mutations, one of the genetic factors involved in DCIS progression [41,42]. Paradoxically, the previous reports, combined with our results, suggest that the transition to *PGR*-positive DCIS may slow cancer progression. The thin edge from *PGR*-positive DCIS to other clusters in the PAGA graph also supports this hypothesis.

The *spatial stratification* algorithm (Output 4) extended the spatial domain to encompass not only the inner section but also the surrounding area (Figs 2, 5, and 6). Our results showed that *CD8A* was spatially localized in closer proximity to DCIS regions and more distally to IDC regions (Fig 6). In the previous report, a reduction in the number of activated CD8+ T cells was observed in IDC than those in DCIS [43]. Our results are consistent with this previous report and support the validity of the SKNY analysis. Furthermore, by extracting gene sets with high correlation to *CD8A* expression in the stratified sections, SKNY evaluated biochemical pathways and cell-cell interactions within the regions where CD8+ T cells were present. Although the present study focused on a relatively limited number of genes, thereby only detecting typical antigen-presenting pathways, the application of a more comprehensive gene panel could facilitate the identification of specific drug target molecules.

The *detection* algorithm (Output 1) delineated different tumor shapes based on histological features (Fig 2). The enrichment of cancer cells and stromal markers within and outside the spatial domains indicates accurate separation of the tumor and stroma. Myoepithelial cells surround the ductal epithelium for structural support [44], and our results also showed that myoepithelial cell markers, including *ACTA2*, *MYLK*, and *KRT14*, were enriched in the perispatial domain of the tumor, suggesting high-quality detection of tumor contours using our algorithm. This high-quality contour guaranteed subsequent SKNY analyses, including *clustering*, *trajectory estimation*, and *spatial stratification*.

This study had some limitations. First, although our analysis of breast cancer and metastatic colorectal cancer samples confirmed advantages of the SKNY, it is necessary to verify SKNY performance using larger samples. In an integrated analysis of data from 14 metastatic colorectal cancer patients, SKNY demonstrated superior capacity for

microenvironment characterization compared to conventional cell-level analysis methods. Even if the number of genes and samples in the panel increases in the future, SKNY can still catalog the microenvironment appropriately. Second, in the present analysis, the spatial omics data were converted to 10 × 10-μm grids, which may make it difficult to detect thin tissues, such as monolayered epithelium. However, setting the grid data to a smaller size should result in insufficient sensitivity of the marker genes on each grid. Therefore, it is necessary to consider the balance between grid size and marker gene sensitivity for each specimen and gene panel.

## Conclusion

In conclusion, SKNY can be used in microenvironmental analyses to provide valuable insights into its pathological functions. It should be applicable not only to the TME but also to a wide range of microenvironments, such as tertiary lymphoid structures and myocardial and neuronal microenvironments.

## Methods

### Ethics statement

The study protocol was approved by Institutional review board of national cancer center (UMIN000027887).

### Data acquisition and pre-processing

Breast cancer data from Xenium were downloaded from a public repository (https://www.10xgenomics.com/jp/products/xenium-in-situ/preview-dataset-human-breast). The 'ReadXenium' function from stlearn (v0.4.12) was used to read the HE images (https://www.dropbox.com/s/th6tqqgbv27o3fk/CS1384_post-CS0_H%26E_S1A_RGB-shlee-crop.png?dl=1) and files containing gene expression and cell coordinates (Xenium_FFPE_Human_Breast_Cancer_Rep1_cell_feature_matrix.h5 and Xenium_FFPE_Human_Breast_Cancer_Rep1_cells.csv.gz). The 'tl.cci.grid' function in stlearn was used to simplify the coordinate data into grid data ( $Grid^{gene}_{column,row}, gene = \{ABCC11, ACTA2, ACTG2, \ldots, ZNF562\}, column = \{1,2,3,\ldots,752\}, row = \{1,2,3,\ldots,547\}$ ) at the 10-μm interval.

Data for ovarian cancer, colorectal cancer, and melanoma were downloaded from the following URLs: https://www.10xgenomics.com/jp/datasets/ffpe-human-ovarian-cancer-data-with-human-immuno-oncology-profiling-panel-and-custom-add-on-1-standard, https://www.10xgenomics.com/jp/datasets/ffpe-human-colorectal-cancer-data-with-human-immuno-oncology-profiling-panel-and-custom-add-on-1-standard, https://www.10xgenomics.com/jp/datasets/human-skin-data-xenium-human-multi-tissue-and-cancer-panel-1-standard

In the TRIUMPH trial [28], formalin-fixed paraffin-embedded (FFPE) biopsy specimens were collected from 14 patients with HER2-amplified metastatic colorectal cancer. Twenty-four FFPE tissue sections were obtained, representing pre- and post-treatment time points or a single timepoint, depending on the patient's treatment course. Spatial gene expression profiling was performed on the samples using the Xenium platform (10x Genomics, Pleasanton, CA, USA), which enables in situ analysis of RNA expression at subcellular resolution. For our analysis, a custom panel of 300 genes specifically designed for colorectal cancer research was utilized. The Xenium workflow consists of several key steps: tissue permeabilization and pretreatment, hybridization with gene-specific probes, rolling circle amplification (RCA) of target sequences, detection using fluorescently labelled oligonucleotides, and high-resolution imaging and data acquisition.

## Detection of spatial domain

The pre-spatial domain ($S_{pre}$) was determined by subtracting the grids with a negative marker (example: *SFTPB*; breast cancer, melanoma, ovarian cancer, colorectal cancer, and metastatic colorectal cancer: N/A) from the grids with a positive marker (example and breast cancer: *CDH1*; melanoma: *MLANA*; ovarian cancer, colorectal cancer, and metastatic colorectal cancer: *EPCAM*) (S11A Fig). The SKNY program can detect pre-spatial domains based on user selection. For example, if a user wants to obtain only the tumor region without the normal epithelium, logical subtraction between a positive marker's expression (e.g., *CDH1*) and a negative marker (e.g., *SFTPB*) can be performed.

$$S_{pre} = \left( Expr(Grid_{column,row}^{positive\,marker}) > 0 \right) - \left( Expr\left(Grid_{column,row}^{negative\,marker}\right) > 0 \right)$$

where *Expr* is defined as a function of extracting gene expression counts from the grid (S11A Fig). To remove noise from the pre-spatial domain, the "medianBlur" function (kernel size: 3×3) from the Python library opencv (v4.8.1) was applied, resulting in the formation of a denoised spatial domain (S11B Fig).

The STAGATE algorithm [24] was also used to extract spatial domain clusters for comparison with the existing methods. To annotate the extracted spatial domain clusters, the expression levels of epithelial markers (*CDH1*, *EPCAM*) were compared, and cluster 1, 3, and 9, which showed overexpression, was extracted as the spatial domain of the tumor. To assess the concordance between SKNY and STAGATE in the spatial domains, the Jaccard coefficient, which indicates the percentage of agreement between each lattice, was calculated.

## Measurement of distance from the boundary line of the spatial domain

The boundary line was identified using the 'findContours' function from opencv in the spatial domain (S11B Fig). All adjacent grids were connected by edges and weighted according to the Euclidean distance: 1 for vertical and horizontal edges and a root of 2 for diagonal edges (S12 Fig). The shortest path from the boundary line to the other grids was measured using the multi-source Dijkstra method [45] to determine the distance from the spatial domain edges.

## Segmentation from a spatial domain to individual spatial domains

The function 'connectedComponentsWithStats' from opencv was used to divide the entire image of the spatial domain into individual spatial domain ($S_d$, $d = \{1,2,3,\ldots,426\}$). The gene expression within each spatial domain was averaged.

## Spatial stratification by spatial domains

Using the measured distances (S12 Fig), a stratification was performed with a half-open interval of 30 μm to determine the partial area ($P$) as a stratified spatial domain ($P_{x<\mu\leq x+30}, x = \{-120,-90,-60,\ldots,150\}$) (S11B Fig). For the stratified spatial domain of the outer section of the spatial domain ($P_{x<\mu\leq x+30}, x = \{0,30,60,\ldots,150\}$), the rectangle that enclosed each $S_d$ was then extracted, and each rectangle was enlarged by *x* μm to produce a rectangle including each stratified spatial domain ($R_{d,x+30}, d = \{1,2,3,\ldots,426\}, x = \{0,30,60,\ldots,150\}$). The stratified spatial domain exclusive to the others ($PS_{d,x<\mu\leq x+30}, d = \{1,2,3,\ldots,426\}, x = \{0,30,60,\ldots,150\}$) was calculated as follows:

$$PS_{d,\,x<\mu\leq x+30} = P_{x<\mu\leq x+30} \wedge R_{d,x+30} \wedge \neg \bigcup_{0\leq i<j\leq n} \left( R_{i,x+30} \wedge R_{j,x+30} \right)$$

where $\wedge$ represents the product sum, and $\cup$ represents the union set. The gene expression of each $PS_{d,\, x < \mu \leq x+30}$ was defined as the average gene expression of the grids within it. For simplicity, a flow assuming three adjacent spatial domains (S1, S2, and S3) and their (0, 30] stratified spatial domains ($P_{0 < \mu \leq 30}$) is shown (S11C Fig). First, the rectangles covering the perimeter of each of S1, S2, and S3 are extracted and expanded by 30 μm ($R_{1,30}$, $R_{2,30}$, and $R_{3,30}$), and their pairwise intersection ($\bigcup_{0 \leq i < j \leq n}\left(R_{i,x+30} \wedge R_{j,\, x+30}\right)$) was taken. Then, to define stratified spatial domains specific to each spatial domain ($PS_{1,\, 0<\mu\leq30}$, $PS_{2,\, 0<\mu\leq30}$, and $PS_{3,\, 0<\mu\leq30}$), intersection sets of the rectangle ($R_1$, $R_2$, or $R_3$), and complement of the pairwise intersection ($\neg\bigcup_{0 \leq i < j \leq n}\left(R_{i,x+30} \wedge R_{j,\, x+30}\right)$) were taken.

## Diversity analysis in the spatial domain

To compare the alpha diversity of gene expression between the segmented spatial domains and previously annotated cancer cells [26], the 'diversity.alpha.chao1', 'diversity.alpha.shannon', and `diversity.alpha.observed_otus` in the Python library scikit-bio was used to calculate Chao1, Shannon, and observed features, respectively [46].

## Clustering of the spatial domains

The 'pp.log1p' function from scanpy (v 1.9.8) was used to log-transform gene expression in each spatial domain ($S_d$). Subsequently, the 'pp.pca' function was used for dimension reduction through PCA. Fifty principal components were extracted in the order of highest eigenvector. The 'pp.neighbours' and 'tl.leiden' functions from the scanpy were adapted to form spatial domain clusters for leiden clustering. The 'tl.umap' function was used to place leiden embeddings on the UMAP two-dimensional space.

## Trajectory estimation of the spatial domains

For trajectory inference by the PAGA algorithm [15], the "tl.paga" function of scanpy was used to construct the neighborhood graph of the spatial domain cluster, followed by estimation of the pseudotime by adapting the "tl.dpt" function. R libraries of Monocle [27] and Slingshot [17] were also used to confirm trajectory estimated by PAGA.

## Statistical analysis

Pearson's product-moment correlation coefficient was used to analyze the correlation between pseudotime and gene expression. Welch's t-test was used to compare alpha diversity between the two groups. The Kruskal-Wallis test was used to compare gene expression between multiple groups. The "GLM" function in the Python library statsmodels (v0.14.1) was used to construct the general linearized model. KEGG pathway analysis was conducted using the ShinyGO (v0.80) [47], with an False Discovery Rate cutoff of 0.001.

## Visualization

Xenium Explororor (v1.3) or the "pl.gene_plot" function in stlearn was used for the visualization of the Xenium data.

## Supporting information

**S1 Fig. Annotation of the spatial domain using the STAGATE algorithm.** Spatial distribution of each cluster by STAGATE algorithm at (A) single-cell level and (B) grid level. (C) Dotplot showing markers of cell types and expression patterns of genes associated with tumor subtypes. Clusters 1, 3, and 9 correspond to the tumor spatial domain.
(TIF)

**S2 Fig. Detection of tumor spatial domain using SKNY for multiple cancer types.**
H&E staining images and detected spatial domains of ovarian cancer, colorectal cancer, and melanoma. The yellow and green colors indicate spatial domains and the boundary, respectively.
(TIF)

**S3 Fig. Measurement of distance from the surface of spatial domains.** (A) Heatmap indicating distance from surfaces of spatial domains. (B) The red contour lines indicate distance from the surface of spatial domains at the 30-μm interval.
(TIF)

**S4 Fig. Comparison of alpha-diversity index based on gene expression.** (A) Box plot of alpha-diversity index (Chao1) of expression between cancer cells and spatial domains. Box plot of (B) Chao1, (C) observed features, and (D) Shannon of expression among cells annotated by Janesick *et al.* and spatial domains.
(TIF)

**S5 Fig. Trajectory analysis of spatial domains using Monocle and Slingshot algorithms and autocorrelation analysis of marker gene for endothelial cell.** (A) The plot shows the Uniform Manifold Approximation and Projection (UMAP) representation of spatial domains, colored by their Leiden cluster assignments, overlaid with the trajectories predicted by the Monocle algorithm. The root point of the trajectory is labeled by the white node, and the branch points are labeled by the gray node. (B) The plot shows the UMAP representation of spatial domains, overlaid by their Leiden cluster assignments (left), overlaid with the trajectories predicted by the Slingshot algorithm (right). (C) Line plot depicting the relationship between Leiden clusters and Geary's C for the genes *PECAM1* (blue), *VWF* (orange), and *CD93* (green).
(TIF)

**S6 Fig. Trajectory analysis illustrating the progression flow for the tumor microenvironment.** (A) Violin plots showing marker expressions of *PECAM1*, *CD68*, *MMP2*, *CXCR4*, and *CXCL12* on the estimated trajectory path (IDC path #1) in the (−30, 0], (0, +30], and (+30, +60] sections. The color scale indicates the mean of pseudotimes in each cluster. The annotated values represent *P* values of the significance test. Red dots in the Figs indicate the mean the gene expression level. (B) Summary of the gene expression dynamics. Red and blue colors indicate the overrepresentation and underrepresentation of the gene expressions.
(TIF)

**S7 Fig. Identification of gene clusters correlated with CD8A distribution in DCIS and IDC.** (A) Top 10 genes with correlation coefficients with *CD8A* of DCIS or IDC. (B) Bar plot shows the expression density of *CD8A*, *CCL5*, and *CX3CR1* stratified by section. The red series indicates DCIS, and the blue series indicates IDC. The respective color gradients indicate each stratified interval. Error bars represent 95% confidence intervals (CI). (C) Spatial expression distribution of *CD8A*, *CCL5*, and *CX3CR1* in each ROI. (D) Lollipop plot shows significantly enriched KEGG pathways. The x-axis denotes fold enrichment, plot size represents the number of genes, and the color represents −log10(FDR).
(TIF)

**S8 Fig. SKNY application to spatial domain detection in the Xenium data set from the TRIUMP trial.** Spatial domains of metastatic colorectal cancer. The yellow and green colors indicate spatial domains and the boundary, respectively.
(TIF)

**S9 Fig. Clustering at the cellular level and extraction of tumor cell clusters.** (A) t-Distributed stochastic neighbor embedding (tSNE) plot of 1,133,514 cancer cells. The plot is colored according to the clusters determined by the leiden algorithm. (B) tSNE from A colored by *EPCAM* expression. (C) Extraction of clusters with high *EPCAM* and rearrangement of tSNE space.
(TIF)

**S10 Fig. Expression distribution of fibroblast markers and cytokines between spatial domain-based and tumor cell-based tSNE spaces. t-Distributed stochastic neighbor embedding (tSNE) plot of spatial domain and cancer cells.** The plots are colored according to the expression of *COL1A1*, *FGFR2*, *CXCL12*, and *CXCL13*.
(TIF)

**S11 Fig. Workflow of SpatialKNifeY algorithm.** (A) The schematic illustrates the process of identifying pre-spatial domains that express positive markers and do not express negative markers within a $10 \times 10$ μm grid. (B) The schematic shows the workflow for identifying and stratifying the spatial domain using three sequential steps: median filtering, contouring, and measurement of distance. (C) The schematic demonstrates the procedure for extracting stratified spatial domains that do not overlap with other ones.
(TIF)

**S12 Fig. Measurement of shortest distance from contour of spatial domains.** Left: Representation of a graph with weighted edges, where nodes are categorized as Outside the spatial domain (black), contour (green), and inside the spatial domain (yellow). Right: The result of applying the Multi-Source Dijkstra Algorithm is to compute the shortest distance from the contour. The calculated distances for each node are indicated in red.
(TIF)

## Author contributions

**Conceptualization:** Shunsuke A Sakai, Shun-Ichiro Kageyama, Riu Yamashita.

**Data curation:** Mitsuho Imai, Yoshiaki Nakamura, Takayuki Yoshino.

**Formal analysis:** Shunsuke A Sakai.

**Funding acquisition:** Shunsuke A Sakai.

**Investigation:** Shunsuke A Sakai, Ryosuke Nomura, Satoi Nagasawa, SungGi Chi, Ayako Suzuki, Yutaka Suzuki, Mitsuho Imai, Yoshiaki Nakamura, Takayuki Yoshino.

**Methodology:** Shunsuke A Sakai, Ryosuke Nomura, Satoi Nagasawa, SungGi Chi, Ayako Suzuki, Yutaka Suzuki, Shumpei Ishikawa, Riu Yamashita.

**Project administration:** Shunsuke A Sakai.

**Resources:** Mitsuho Imai, Yoshiaki Nakamura, Takayuki Yoshino.

**Software:** Shunsuke A Sakai.

**Supervision:** Katsuya Tsuchihara, Shun-Ichiro Kageyama, Riu Yamashita.

**Validation:** Shunsuke A Sakai, Ryosuke Nomura.

**Visualization:** Shunsuke A Sakai, Ryosuke Nomura.

**Writing – original draft:** Shunsuke A Sakai.

**Writing – review & editing:** Shunsuke A Sakai, Ryosuke Nomura, Satoi Nagasawa, SungGi Chi, Ayako Suzuki, Yutaka Suzuki, Mitsuho Imai, Yoshiaki Nakamura, Takayuki Yoshino, Shumpei Ishikawa, Katsuya Tsuchihara, Shun-Ichiro Kageyama, Riu Yamashita.

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
