## [Decision Letter · Decision Letter 0]

21 Nov 2024

PCOMPBIOL-D-24-01692SpatialKNifeY (SKNY): Extending from spatial domain to surrounding area to identify microenvironment features with single-cell spatial omics dataPLOS Computational BiologyDear Dr. Yamashita, Thank you for submitting your manuscript to PLOS Computational Biology. After careful consideration, we feel that it has merit but does not fully meet PLOS Computational Biology's publication criteria as it currently stands. Therefore, we invite you to submit a revised version of the manuscript that addresses the points raised during the review process. Please submit your revised manuscript within 30 days Jan 21 2025 11:59PM. If you will need more time than this to complete your revisions, please reply to this message or contact the journal office at ploscompbiol@plos.org. Please include the following items when submitting your revised manuscript: * A rebuttal letter that responds to each point raised by the editor and reviewer(s). You should upload this letter as a separate file labeled 'Response to Reviewers'. This file does not need to include responses to formatting updates and technical items listed in the 'Journal Requirements' section below. * A marked-up copy of your manuscript that highlights changes made to the original version. You should upload this as a separate file labeled 'Revised Manuscript with Track Changes'. * An unmarked version of your revised paper without tracked changes. You should upload this as a separate file labeled 'Manuscript'. If you would like to make changes to your financial disclosure, competing interests statement, or data availability statement, please make these updates within the submission form at the time of resubmission. Guidelines for resubmitting your figure files are available below the reviewer comments at the end of this letter. We look forward to receiving your revised manuscript. Kind regards,Wei Li, Ph.D.Academic EditorPLOS Computational BiologyMarc BirtwistleSection EditorPLOS Computational Biology

Feilim Mac Gabhann

Editor-in-Chief

PLOS Computational Biology

Jason Papin

Editor-in-Chief

PLOS Computational Biology

**Journal Requirements:**

2) Please provide an Author Summary. This should appear in your manuscript between the Abstract (if applicable) and the Introduction, and should be 150u2013200 words long. The aim should be to make your findings accessible to a wide audience that includes both scientists and non-scientists. Sample summaries can be found on our website under Submission Guidelines:

3) Your manuscript is missing the following sections: Introduction.  Please ensure all required sections are present and in the correct order. Make sure section heading levels are clearly indicated in the manuscript text, and limit sub-sections to 3 heading levels. An outline of the required sections can be consulted in our submission guidelines here:

5) We have noticed that you have uploaded Supporting Information files, but you have not included a list of legends. Please add a full list of legends for your Supporting Information files after the references list.

6) We notice that your supplementary Figures are included in the manuscript file. Please remove them and upload them with the file type 'Supporting Information'. Please ensure that each Supporting Information file has a legend listed in the manuscript after the references list.

7) Some material included in your submission may be copyrighted. According to PLOSu2019s copyright policy, authors who use figures or other material (e.g., graphics, clipart, maps) from another author or copyright holder must demonstrate or obtain permission to publish this material under the Creative Commons Attribution 4.0 International (CC BY 4.0) License used by PLOS journals. Please closely review the details of PLOSu2019s copyright requirements here: PLOS Licenses and Copyright. If you need to request permissions from a copyright holder, you may use PLOS's Copyright Content Permission form.

Potential Copyright Issues:

i) The following Figure contains a logo or branding: Figure 1B. We are not permitted to publish this under our CC-BY 4.0 license, even with permission. We ask that you please remove or replace it.

8) Please amend your detailed Financial Disclosure statement. This is published with the article. It must therefore be completed in full sentences and contain the exact wording you wish to be published.

**Reviewers' comments:**

Reviewer's Responses to Questions

**Comments to the Authors:**

Reviewer #1: The manuscript presents a toolkit that combines spatial omics data with histological image processing to study tumor microenvironments. With previously published breast and colorectal cancer datasets, SKNY detects spatial domains reflecting tumor regions, clusters these domains by malignancy, and extends analysis into the surrounding TME to assess cellular diversity and tumor progression trajectories. The study claims consistency with known cancer progression mechanisms and suggests SKNY’s capability for integrated analysis across patient samples to catalog TME features critical to disease progression. In my opinion, this manuscript is suited to be published in PLOS Computational Biology after addressing the following comments.

1. In the introduction, it'd be helpful to elaborate on existing microenvironmental analysis tools in the context of spatial omics by pointing out their limitations and how SKNY solve them.

2. The manuscript highlights increased diversity within spatial domains compared to isolated cancer cells, indicating the presence of various cell types within the TME. However, it lacks sufficient statistical explanation for how the α-diversity index directly correlates with spatial heterogeneity. A comparative analysis using other diversity indices or validation against independently annotated datasets would enhance credibility.

3. Clustering was conducted using PCA and UMAP, which provided interpretative 2D visualizations, yet no justification was given for the number of clusters. It would be beneficial to include a validation step, like silhouette analysis, to confirm the chosen number of clusters.

4. The study uses PAGA to model DCIS-to-IDC progression, but it could benefit from a pseudotime validation approach, such as Slingshot or Monocle, to ensure robust trajectory construction. Testing multiple trajectory algorithms would allow better triangulation of findings and validation of SKNY’s approach for trajectory estimation.

5. While significant associations between endothelial infiltration and cancer stages were identified, certain analyses lack sufficient statistical controls. The Kruskal-Wallis test is appropriate but limited for stratified spatial data. Applying mixed-effect models to account for intra-sample variability or leveraging spatial autocorrelation metrics could refine these results.

Reviewer #2: This paper presents SpatialKNifeY, that works on spatial transcriptomics data from Xenium platform. The tool can detect spatial domains and associated microenvironments. It can also provide trajectory estimation. The tool has been showcased on several datasets.

The code is well organized and documented in the Github repository. I have the following comments on the manuscript

Line 68: Please use full form before using acronym

I found the method difficult to understand (likely because of my confusion, as can be seen in comments that follow).

What is meant by pre-spatial domain? The authors need to clarify this.

Line 491: What is meant by pre-spatial domain?

Line 499: I did not understand this equation. Perhaps, description will be helpful.

The STAGATE algorithm is misspelled as STARGATE in some places.

Line 513: What is meant by domain surface?

Line 514: What is an edge grid? Due to lack of my familiarity with this term, I was not able to follow this entire paragraph.

Line 516: it should be square root of 2, not 2. [It is correctly mentioned in the referred figure, however].

Line 522: It is not clear how connectedComponentsWithStats method from opencv could divide a spatial domain into individual spatial domains. The input needs to be an image. Do you pass in the entire image, or only a portion capturing the spatial domain?

I could not understand the notation in line 528, what is meant by P ?

I could not understand the equation in line 534. Why are there two union symbols? A written description would be beneficial here.

Line 554: typo snappy >> scanpy

A flowchart of the method may be helpful.

**Have the authors made all data and (if applicable) computational code underlying the findings in their manuscript fully available?**

Reviewer #1: Yes

Reviewer #2: Yes

PLOS authors have the option to publish the peer review history of their article (what does this mean? ). If published, this will include your full peer review and any attached files.

**Do you want your identity to be public for this peer review?** For information about this choice, including consent withdrawal, please see our Privacy Policy .

Reviewer #1: No

Reviewer #2: **Yes: ** M Saifur Rahman

**Figure resubmission:**
---

## [Decision Letter · Decision Letter 1]

3 Feb 2025

Dear Unit leader Yamashita,

We are pleased to inform you that your manuscript 'SpatialKNifeY (SKNY): Extending from spatial domain to surrounding area to identify microenvironment features with single-cell spatial omics data' has been provisionally accepted for publication in PLOS Computational Biology.

Best regards,

Wei Li, Ph.D.

Academic Editor

PLOS Computational Biology

Marc Birtwistle

Section Editor

PLOS Computational Biology

Reviewer's Responses to Questions

**Comments to the Authors:**

Reviewer #1: The authors have provided a comprehensive revision of the manuscript. The responses and changes implemented in the revised manuscript were well-articulated and adequately addressed the reviewers' comments.

Reviewer #2: All my comments have been addressed. I thank the authors for their effort. I am happy to recommend publication of this paper.

**Have the authors made all data and (if applicable) computational code underlying the findings in their manuscript fully available?**

Reviewer #1: Yes

Reviewer #2: None

PLOS authors have the option to publish the peer review history of their article (what does this mean? ). If published, this will include your full peer review and any attached files.

**Do you want your identity to be public for this peer review?** For information about this choice, including consent withdrawal, please see our Privacy Policy .

Reviewer #1: No

Reviewer #2: **Yes: ** M Saifur Rahman

---

## [Editor Report · Acceptance letter]

PCOMPBIOL-D-24-01692R1

SpatialKNifeY (SKNY): Extending from spatial domain to surrounding area to identify microenvironment features with single-cell spatial omics data

Dear Dr Yamashita,

I am pleased to inform you that your manuscript has been formally accepted for publication in PLOS Computational Biology. Your manuscript is now with our production department and you will be notified of the publication date in due course.

With kind regards,

Zsofia Freund
